# *Weizmannia coagulans* BC99 Relieves Constipation Symptoms by Regulating Inflammatory, Neurotransmitter, and Lipid Metabolic Pathways: A Randomized, Double-Blind, Placebo-Controlled Trial

**DOI:** 10.3390/foods14040654

**Published:** 2025-02-15

**Authors:** Qiuxia Fan, Yinyin Gao, Yiqing Zhou, Jinghui Wu, Haotian Wang, Yao Dong, Zhonghui Gai, Ying Wu, Shuguang Fang, Shaobin Gu

**Affiliations:** 1College of Food and Bioengineering, Henan University of Science and Technology, Luoyang 471000, China; 13279322787@163.com (Q.F.); gaoyinyin1999@163.com (Y.G.); 15036058770@163.com (Y.Z.); wujinghui2024@163.com (J.W.); wanghaotian268@163.com (H.W.); wuying2000@126.com (Y.W.); 2Henan Engineering Research Center of Food Material, Henan University of Science and Technology, Luoyang 471023, China; 3Henan Engineering Research Center of Food Microbiology, Luoyang 471000, China; 4Department of Research and Development, Wecare Probiotics Co., Ltd., Suzhou 215200, China; florady0327@163.com (Y.D.); zhgai@aliyun.com (Z.G.); frank.fang@wecare-bio.com (S.F.); 5Germline Stem Cells and Microenvironment Lab, College of Animal Science and Technology, Nanjing Agricultural University, Nanjing 210095, China

**Keywords:** *Weizmannia coagulans* BC99, constipation, neurotransmitters, inflammation, serum metabolism

## Abstract

Probiotics have attracted increasing attention due to their benefits in terms of relieving gastrointestinal ailments, including constipation. This study evaluated the potential of *Weizmannia coagulans* BC99 for clinical remission of constipation in adults. In this randomized, double-blind, and placebo-controlled trial, 90 individuals with constipation were divided between a BC99 and a placebo group for an 8-week intervention duration. The spontaneous bowel movement (SBM) frequency, patient assessment of constipation symptoms (PAC-SYM), patient assessment of constipation quality of life (PAC-QOL), inflammatory cytokines, neurotransmitters, and serum metabolites were investigated before and after the intervention. The results showed that BC99 intervention significantly improved constipation symptoms and quality of life in adults with constipation, as evidenced by an increased SBM score and decreased PAC-SYM and PAC-QOL scores. Additionally, BC99 supplementation increased the levels of neurotransmitters (5-HT, MTL, AChE, and BDNF) associated with intestinal motility and alleviated inflammation in participants with constipation, as supported by higher levels of anti-inflammatory factors (IL-4, IL-10) and lower levels of pro-inflammatory factors (IL-6, IFN-γ) in the BC99 group. Furthermore, BC99 altered the abundance of 93 metabolites and affected biological pathways correlated with gastrointestinal motility, including sphingolipid metabolism, steroid hormone biosynthesis, and glycerophospholipid metabolism. This study demonstrates the effectiveness of the *W. coagulans* BC99 strain in relieving constipation in adults, and reveals its potential mechanism of action. These findings provide a scientific basis for BC99 as an effective and safe probiotic for constipation treatment.

## 1. Introduction

Constipation is a common and complex gastrointestinal disorder besetting about 15% of the world’s population [1], which is characterized by rough, dry, or hard stool, and difficulty in or a reduced frequency of bowel movements [2]. In most cases, constipation is accompanied by great discomfort, loss of appetite, and abdominal pain and distension [3,4]. In addition, sleep disturbances and anal fissures are also frequently observed in patients with constipation, finally resulting in a decline in or loss of quality of life [5]. Importantly, it is also associated with other intestinal diseases, metabolic diseases, cardiovascular disease, and neurological disease, exerting negative impacts on human health [6,7]. Currently, the management of constipation poses a challenge. Conventional pharmacological interventions, such as osmotic laxatives, stool softeners, and stimulant laxatives, may not consistently provide adequate relief [8], and excessive use of these treatments could lead to adverse effects [9]. Therefore, there is an urgent need for safe and efficient treatments for constipation.

In recent years, probiotics have been demonstrated to have great prospects in the treatment and prevention of gastrointestinal diseases, including constipation. The intestinal microbiota affects gastrointestinal health through a variety of mechanisms, including regulating the immune system, maintaining intestinal barrier function, and secreting metabolites. These processes directly or indirectly affect the occurrence and development of gastrointestinal diseases [10]. There is a two-way communication network between the intestinal microbiota and the brain, the so-called “gut–brain axis”, which regulates emotions, cognitive function, and overall health status through pathways such as neurotransmitters, hormones, and immune signals [11]. Many studies have shown that probiotic intervention can effectively alleviate constipation and associated symptoms with few adverse reactions [12,13], based on randomized trials for adults and older adults. The underlying mechanisms by which probiotics relieve constipation may differ between different types of probiotics, but they mainly involve regulating gut microbiota composition, enhancing intestinal motility, improving the intestinal environment, and affecting levels of neurotransmitters and inflammation [14]. *Weizmannia coagulans* (*W. coagulans*) BC99 is a Gram-positive, facultatively anaerobic, and spore-producing bacterium with high biosafety [15,16,17]. Compared with lactic acid bacteria, *W. coagulans* BC99 has the advantages of outstanding adaptability, heat resistance, vigorous metabolism, and high revival rates [18]. *W. coagulans* has multiple benefits for human intestinal health, including regulating intestinal flora balance, restoring intestinal motility, and inhibiting pathogen growth and adhesion, as well as producing short-chain fatty acids [19]. In previous double-blind, placebo-controlled studies, the effectiveness of *W. coagulans* UniqueIS2 and BC99 in the treatment of constipation in adults has been demonstrated [20,21]. However, more in-depth investigations are required to fully reveal the underlying mechanisms of *W. coagulans* against constipation problems.

Metabolomics is an important approach to monitoring metabolic changes and differentiating between different patient populations. Studies have found that metabolic disorders are closely related to the occurrence of constipation, because constipation is not only a local intestinal problem, but also a complex disease that is closely related to the whole body’s metabolic state [22]. Therefore, it is of great significance and need to uncover the alternations in metabolic profiles caused by intervention with *W. coagulans* BC99, which will contribute to a better understanding of the mode of action of *W. coagulans* BC99 for constipation relief in adults. This present study aims to systematically evaluate the impacts of an 8-week intervention with *W. coagulans* BC99 on constipation symptoms, quality of life, inflammatory cytokines, and neurotransmitter levels, as well as serum metabolic profiles, in participants with constipation, through a randomized, double-blind, and placebo-controlled trial, with maltodextrin as the placebo and BC99 (2 Billion CFU/day) as the intervention. The results of our study will provide a necessary scientific basis for the application of *W. coagulans* BC99 in the treatment of constipation in adults.

## 2. Materials and Methods

### 2.1. Study Design

This research was a randomized, double-blind, placebo-controlled study, which received approval from the Ethics Commission of the First Affiliated Hospital of Henan University of Science and Technology at 26 April 2024 (Ethical approval number 2024-03-K0054), Clinical trial registration number is NCT06637397 (https://clinicaltrials.gov/, accessed on 26 April 2024). The present study was conducted from July 2024 to October 2024 at the College of Food and Bioengineering, Henan University of Science and Technology, in accordance with the tenets of the Declaration of Helsinki. Participants were recruited through bulletin boards and online social networking sites at Henan University of Science and Technology. A preliminary phone interview was carried out to verify the suitability of candidates based on the criteria for inclusion and exclusion. Following a screening process involving identity authentication and physical examination, a total of 100 individuals were deemed eligible for participation. Prior to enrollment, all participants received a detailed description of the study protocol and signed a written informed consent form.

### 2.2. Inclusion/Exclusion Criteria

The inclusion criteria were as follows: (1) Chinese adults who met the Rome III diagnostic criteria and were aged ≥20 years; (2) patients with chronic constipation (duration of more than 6 months, with less than 3 bowel movements per week and/or Bristol Scale Type 1 and 2); (3) patients in whom loose stools were rare without the use of laxatives; (4) The criteria for irritable bowel syndrome were insufficient; (5) patients that could complete the study according to the requirements of the trial protocol; (6) patients that had signed the informed consent form; (7) the study participants (including male subjects) had no fertility plans within 14 days before screening, and would voluntarily take effective contraceptive measures within 6 months after the end of the trial. Only those who met all of the above conditions could be selected.

The exclusion criteria were as follows: (1) Those who took items with similar functions to the test subjects in the short term, which may have affected their judgment of the results; (2) those who regularly used drugs that affect bowel habits; (3) patients who changed their diet during the study (such as regularly using a high-fiber diet, according to the recommended food score (RFS)); (4) those who were allergic to probiotics or the ingredients used in this study, those with severe immunodeficiency, or those with serious psychological or psychiatric diseases; (5) women who were pregnant, breastfeeding, or planning to become pregnant; (6) those who suffered from diseases that may cause secondary constipation, such as diabetes, thyroid disease, Alzheimer’s disease, severe metabolic diseases, malignant tumors, and severe lesions of important organs (cardiovascular, lung, liver, kidney, etc.); (7) those who had used antibiotics in the two weeks prior; (8) those who did not consume the test samples as required, or were not followed up on time, resulting in uncertainty of efficacy; (9) those who had participated in another clinical trial within one month before enrollment; (10) subjects who were judged by other researchers to be unsuitable for participation. Those who met any of the above conditions were not selected.

### 2.3. Sample Size and Randomization

As presented in Figure 1, the experiment was performed in a 1:1 ratio, with randomization to the probiotic group (n = 45) or the placebo group (n = 45). The randomization was carried out utilizing a computer-generated randomization schedule with a block size of 4, resulting in six potential blocks with diverse sequences (AABB, ABBA, ABAB, BBAA, BAAB, BABA). This method guaranteed an equal distribution of participants between the two groups over the duration of the study. Participants in the probiotic group received BC99 (1 g/day (2 Billion CFU/day), Wecare Probiotics Co., Ltd., Suzhou, China), and those in the placebo group consumed maltodextrin (1 g/day), for an 8-week period. Patients are advised to take it with warm water (40 °C) after meals. Participants were asked to limit their intake of additional dietary fiber supplements and other functional foods during the study period. The remaining products and related empty packing boxes were recovered.

### 2.4. Primary Outcomes and Secondary Outcomes

The primary outcomes were changes in patient assessment of constipation symptom (PAC-SYM) scores and spontaneous bowel movement (SBM) scores. Secondary outcomes included patient assessment of constipation quality of life (PAC-QOL) score, levels of serum serotonin (5-HT), motilin (MTL), acetylcholinesterase (AChE), brain-derived neurotrophic factor (BDNF), and inflammatory factors (IL-4, IL-6, IL-10, IFN-γ), and serum metabolic profile. A kit manufactured by Shanghai Hepai Biotechnology Co., Ltd. (Shanghai, China) was used, and an assay was performed using a microplate reader (Thermo Fisher Scientific, Waltham, MA USA).

### 2.5. Blood Biochemical Measurements at Baseline

Blood samples from the placebo and BC99 groups were collected following an overnight fast of 12 h, and used for complete testing. For biochemical tests, the collected blood samples were centrifuged at 4 °C and 3000× *g* for 10 min, and the resulting plasma was harvested and stored at −80 °C prior to analysis. An automatic biochemical analyzer (KHB ZY-1280, Shanghai, China) was used to measure liver function, renal function, and other biochemical parameters.

### 2.6. Constipation Symptoms Assessment

Participants were required to complete three questionnaires associated with constipation symptoms during the 8-week BC99 intervention period, including spontaneous bowel movements (SBMs), patient assessment of constipation symptoms (PAC-SYM), and patient assessment of constipation quality of life (PAC-QOL). SBMs are summary values of spontaneous bowel movements associated with complete emptying [23]. The PAC-SYM questionnaire is the standard method for evaluating patient response outcomes in clinical trials of constipation [24]. The PAC-QOL questionnaire is a commonly used indicator to reflect the life quality of participants with constipation, and higher PAC-QOL scores indicate poorer quality of life [24].

### 2.7. Determination of Inflammatory Cytokines and Neurotransmitter Levels

Blood samples from the placebo and BC99 groups were collected before and after the 8-week intervention, followed by immediate centrifugation at 4 °C and 3000× *g* for 15 min. The resulting plasma was harvested and stored at −80 °C prior to analysis. The levels of inflammatory cytokines (IL-6, IL-4, IL-10, and IFN-γ) and neurotransmitters (5-HT, MTL, AChE, and BDNF) were detected using corresponding ELISA kits (Hepeng Biotechnology Co., Ltd., Shanghai, China), according to the manufacturer’s instructions.

### 2.8. Serum Metabolomic Analysis

Blood samples were taken from patients in the placebo and BC99 groups after fasting overnight for at least 12 h, and then centrifuged (15 min, 3000× *g*) to separate serum samples at 0 and 8 weeks. Serum samples (100 µL) were mixed with extraction solution and deuterated internal standards. After vortex (30 s) and sonication (10 min), the resulting mixtures were cooled down, and the precipitated proteins were incubated at −40 °C for one hour. Another centrifugation (4 °C, 13,800× *g*, 15 min) was performed, and the supernatant was collected in a new glass vial for analysis. To ensure consistency in the analysis, equal amounts of supernatant were combined to create a quality control (QC) sample. The QC samples were processed and analyzed in the same fashion as the analytical samples.

An UHPLC system featuring an ACQUITY UPLC BEH Amide column (2.1 mm 50 mm i.d., 1.7 m; Waters) and connected to an Orbitrap Exploris 120 mass spectrometer LC-MSMS was employed for chromatographic separation of the metabolites. In the mobile phase, ammonium acetate (25 mM) and ammonium hydroxide (25 mM) were dissolved in water with a final pH value of 9.75, and acetonitrile acted as the solvent. The MS spectrum was conducted on an Orbitrap Exploris 120 mass spectrometer in IDA mode, using Xcalibur software (version 18.0.1). The parameters were set as follows: the instrument settings included a full mass spectrometry (MS) resolution of 60,000 and a tandem mass spectrometry (MSMS) resolution of 15,000. The sheath gas flow rate was set at 50 arb, while the auxiliary gas flow rate was 15 arb. The capillary temperature was maintained at 320 °C. The ion-spray voltage was set at 3800 V for the positive mode and 3400 V for the negative mode. The normalized collision energy was adjusted to specific values of 20, 30, and 40 electron volts (eV) for the selected reaction monitoring (SRM) experiments. The raw data were ultimately imported into ProteoWizard for conversion to the mzXML format, and data processing was carried out using a proprietary R-based program. R and BiotreeDB (version 3.0) were employed for metabolite identification.

### 2.9. Statistical Analysis

The general principles of statistical analysis were as follows: continuous variables were described using descriptive statistics (number of subjects, mean, standard deviation), while categorical variables were presented as frequencies and percentages. Data analysis was conducted using SPSS 22.0 software. A normality test was performed prior to further analyses. If normality was violated, appropriate transformations or nonparametric tests were applied. The main analysis included paired sample *t*-tests and independent sample *t*-tests to compare within-group and between-group differences pre- and post-intervention. Disparities were reported as the mean ± standard deviation (SD), with *p* < 0.05 considered statistically significant. Principle component analysis (PCA) and orthogonal partial least squares discriminate analysis (OPLS-DA) were carried out on metabolites using SINCA software (version 18.0.1). Pathway analysis was conducted using KEGG (http://www.genome.jp/kegg/, accessed on 23 November 2024) and MetaboAnalyst (http://www.metaboanalyst.ca/, accessed on 23 November 2024). Fisher’s exact test was utilized for KEGG enrichment analysis of differential metabolites.

## 3. Results

### 3.1. Baseline Characteristics

As shown in Appendix A, A total of 100 subjects were recruited for the trial. Of the 50 subjects in each group, 45 completed the study: they were divided between a placebo group (n = 45) and a probiotic group (n = 45). There were 39 women and 6 men in the probiotic group, with a mean age and height of 34.22 ± 3.84 years and 160.25 ± 5.86 cm, respectively. The placebo group consisted of 41 women and 4 men, with a mean age of 35.24 ± 1.61 years and a mean height of 165 ± 5.39 cm. There were no significant differences between the two groups in terms of renal function, liver function, and biochemical parameters (Appendix A). This equivalence between the groups ensured that the observed effects were attributable to the intervention, rather than to demographic or baseline differences.

### 3.2. Effects of BC99 Intervention on Constipation Symptoms and Quality of Life of Patients with Constipation

In order to evaluate whether BC99 supplementation can improve constipation, the spontaneous bowel movements (SBM), patient assessment of constipation symptoms (PAC-SYM), and patient assessment of constipation quality of life (PAC-QOL) scores of the two groups were recorded during the 8-week intervention period. The PAC-QOL questionnaire is a commonly used indicator to reflect the quality of life of participants with constipation, and higher PAC-QOL scores indicate a poorer quality of life. The PAC-SYM score was utilized to access the severity of constipation symptoms. As presented in Figure 2, compared with the placebo group, the average weekly SBM score of the BC99 group significantly increased (*p* < 0.05), indicating the effectiveness of BC99 in increasing the frequency of bowel movements. In addition, the PAC-SYM and PAC-QOL scores of the BC99 group showed significant declines (*p* < 0.05), revealing that BC99 intervention has the function of relieving constipation symptoms and enhancing quality of life.

### 3.3. Effects of BC99 Intervention on Neurotransmitter and Hormone Levels Associated with Gastrointestinal Motility

Neurotransmitters and hormones play important roles in gastrointestinal motility, and their levels are often different between constipated patients and healthy people [25]. According to previous studies, many probiotics alleviate constipation through regulating the levels of these neurotransmitters and hormones associated with gastrointestinal motility [26]. Therefore, changes in the levels of 5-HT, AChE, MTL, and BDNF in the placebo and BC99 groups were measured in order to reveal the possible mechanism of BC99 for constipation relief. It has been reported that the levels of 5-HT, AChE, MTL, and BDNF are positively associated with gastrointestinal motility and constipation improvement. AChE is a key neurotransmitter-metabolizing enzyme that is responsible for breaking down acetylcholine (ACh), thereby regulating intestinal motility. Changes in AChE activity in patients with constipation may affect intestinal motility. For example, some studies have shown that AChE activity is reduced in the intestines of patients with constipation, which may be related to the pathophysiological mechanism of constipation [27]. As illustrated in Figure 3, after the 8-week intervention with BC99, the levels of 5-HT, AChE, MTL, and BDNF experienced significant increases (*p* < 0.05) compared with those of the placebo group, suggesting that BC99 was able to enhance gastrointestinal motility, promote food transport, and thus improve constipation. In addition, it is evident that the levels of 5-HT, MTL, and BDNF in the BC99 group were higher than that at baseline (*p* < 0.01), indicating the effects of BC99 were time-dependent. Furthermore, probiotic intervention can reduce the expression and activity of AChE, possibly through regulating the enteric nervous system. For example, short-chain fatty acids (SCFAs) produced by probiotics can stimulate enteroendocrine cells to release serotonin (5-HT), which is an important neurotransmitter regulating intestinal motility.

### 3.4. Effects of BC99 Intervention on Inflammatory Factor Levels in Participants with Constipation

It has been found that inflammation impairs the integrity of intestinal epithelial cells, finally resulting in reduced gut motility and chronic constipation [26]. Also, serum inflammatory cytokine levels are positively correlated with the severity of constipation [28]. To investigate whether BC99 supplementation could affect the inflammatory response in patients with constipation, the plasma IL-4, IL-6, IFN-γ, and IL-10 levels of the placebo group and the probiotic group were determined in this study. As shown in Figure 4, compared with the placebo group, the levels of anti-inflammatory factors IL-4 and IL-10 significantly increased after 8 weeks of BC99 intervention (*p* < 0.01), while the levels of pro-inflammatory factors IL-6 and IFN-γ significantly declined after the same intervention (*p* < 0.05). The results reveal that BC99 intervention was effective in reducing inflammation levels in patients with constipation, thereby improving constipation symptoms.

### 3.5. Effect of BC99 Intervention on Serum Metabolic Profiles in Patients with Constipation

#### 3.5.1. Identification, Classification, and Validation of Differential Metabolites

In this study, non-targeted metabolomics was employed to uncover the potential mechanism of BC99 in alleviating constipation at the metabolic level. Figure 5A illustrates the use of OPLS-DA to pinpoint significant biomarkers (variable effects on projection, VIP > 1), distinguishing between the BC99 and placebo groups. Separation between samples from the BC99 and placebo groups was observed in the OPLS-DA score plot with R^2^Y = 0.68 and Q2 = −0.69 (Figure 5B). The three-dimensional principal component analysis (Figure 5C) depicts the serum metabolic profiles, with principal components 1, 2, and 3 accounting for 88.9% of the total components. This suggests substantial differences in metabolic profiles between the BC99 and placebo groups. A total of 93 differential metabolites were identified between the two groups, including 82 up-regulated metabolites and 11 down-regulated metabolites (Figure 5D). Appendix A contains detailed information on these important serum metabolites.

This study further analyzed the top 10 differential metabolites that were significantly up- and down-regulated between the BC99 group and the placebo group. These metabolites can be divided into 11 categories, including the following: 1. sphingolipids, 2. organic sulfonic acids and derivatives, 3. steroids and steroid derivatives, 4. benzene and substituted derivatives, 5. fatty acyls, 6. carboxylic acids and derivatives, 7. glycerophospholipids, 8. pyridopyrimidines, 9. piperidines, 10. stilbenes, and 11. organic nitrogen compounds. Among them, the metabolites belonging to sphingolipids showed significant increases after BC99 supplementation, including SM (d18:1/17:0), SM (d18:1/18:0), SM (d18:0/16:0), and palmitoyl sphingomyelin (Figure 6A,B). In contrast, organic heterocyclic compounds (e.g., 9-hydroxyrisperidone and fagomine) and carboxylic acids and their derivatives (e.g., Leu-Leu, Ile-Leu, Leu-Ile, and N-methyltrimethylacetamide) were significantly down-regulated after BC99 intervention (Figure 6C). In addition, the relative abundance of glycerophospholipids (e.g., 1-Palmitoyl-2-docosahexaenoyl-sn-glycero-3-phosphocholine) was significantly increased in the BC99 group (Figure 6B). In conclusion, BC99 intervention has a significant impact on the serum metabolism of participants with chronic constipation.

#### 3.5.2. KEGG Classification and KEGG Enrichment Analysis of Differential Metabolites

The software of MetaboAnalyst 5.0 was used to analyze the metabolic pathways of serum differential metabolites (VIP > 1, *p* < 0.05) between the placebo group and BC99 group. As presented in Figure 7A, these differential metabolites were involved in 15 KEGG pathways, including sulfur metabolism, glycosylphosphatidylinositol (GPI)-anchor biosynthesis, steroid hormone biosynthesis, sphingolipid metabolism, glycerophospholipid metabolism, the sphingolipid signaling pathway, etc. Notably, lipid metabolism included the most differential metabolites, among which steroid hormone biosynthesis included 29.41% of the metabolites, followed by sphingolipid metabolism and glycerophospholipid metabolism, with equal proportions of metabolites (17.65%). KEGG enrichment analysis showed that a total of 15 KEGG pathways were significantly enriched, among which steroid hormone biosynthesis, sphingolipid metabolism, glycerophospholipid metabolism, and the sphingolipid signaling pathway were the four pathways with the most metabolites and the lowest *p*-values (Figure 7B). Collectively, the results highlight the importance of lipid metabolism (especially steroid hormone biosynthesis and sphingolipid metabolism) in the alleviation of constipation mediated by BC99, which is consistent with the changes in levels of neurotransmitters and hormones.

#### 3.5.3. Correlation Analysis Between Key Metabolites and Clinical Indicators

To investigate the potential correlation between key serum metabolites and clinical indicators, correlation analysis was conducted (Figure 8). The top 10 distinct metabolites showing significant up-regulation and down-regulation were employed to elucidate the connection between important markers and evident indicators in individuals with constipation. It is obviously from Figure 8 that SBMs and PCA-QOL show a significant positive correlation with four serum metabolites involved in sphingolipid metabolism, including SM(d18:1/17:0), SM(d18:1/18:0), SM(d17:1/24:1(15Z)), and SM(d18:0/16:0). In addition, these four metabolites are also positively correlated with neurotransmitters (5-HT, AChE, MTL, BDNF) and anti-inflammatory factors (IL-4, IL-10), and negatively correlated with the pro-inflammatory factor IL-6. The findings indicate that these serum metabolites could serve as important indicators that impact clinical variables and disease characteristics via the sphingolipid metabolic pathways.

## 4. Discussion

Constipation is one of the most common gastrointestinal diseases in clinical practice, with a global prevalence of approximately 3% to 21% [29]. There is a diverse range of illnesses whose presence and progression frequently coincide with constipation, such as cardiovascular and cerebrovascular diseases, colon cancer, etc. [6,30]. As individuals become more health-conscious, the utilization of efficient and secure dietary measures to prevent and alleviate constipation becomes a crucial long-term concern. At present, probiotics have become a hot topic as promising and viable alternatives for relieving constipation. Probiotics have been recognized as having potential for relieving constipation, reducing inflammation, and regulating intestinal microbiota disorders and metabolite levels [31,32]; however, their associated mechanisms remain unclear. This study examined the effects of 8-week BC99 treatment on specific biomarkers and differential metabolites associated with constipation relief. There were no significant differences in routine blood levels, liver and kidney function, and physical examination between the two groups at baseline and week 8. In addition, no significant adverse reactions were observed in all patients, confirming the safety of BC99. BC99 intervention was demonstrated to increase bowel movement frequency and quality of life in patients, and relieve constipation symptoms, as supported by the increased SBM score and reduced scores of PAC-SYM and PCA-QOL. Additionally, chronic constipation is often accompanied by inflammatory responses. Inflammatory factors such as TNF-α can increase intestinal permeability, leading to impaired intestinal mucosal barrier function, and subsequently triggering chronic inflammatory responses [33,34]. BC99 may improve intestinal health and metabolic function by regulating the expression of specific cytokines (such as IL-6, TNF-α, etc.). For example, BC99 indirectly affects the secretion of cytokines by regulating the intestinal flora and metabolic pathways (such as SCFA production) [21]. BC99 alleviates dysfunction by increasing the levels of anti-inflammatory cytokines IL-4 and IL-10 and reducing the release of pro-inflammatory factors IL-6 and IFN-γ to normal levels. An important neurotransmitter is 5-HT, mainly released by intestinal eosinophils, which can stimulate intestinal motility and increase intestinal secretion [35]. Studies have shown that 5-HT can promote the contraction of intestinal smooth muscle by binding to 5-HT receptors, thereby accelerating colon transit [36]. AChE activity is usually reduced in patients with constipation, resulting in a decrease in acetylcholine concentration, which weakens the contractile ability of intestinal smooth muscle and further aggravates constipation [27]. MTL is an intestinal motility hormone that can enhance intestinal motility [37]. BDNF secreted by intestinal epithelial cells is a neurotrophic factor that controls colonic peristalsis, and up-regulated levels of BDNF could promote intestinal motility and defecation [38]. This study found that BC99 intervention for 8 weeks can affect intestinal motility by increasing the levels of 5-HT, AChE, MTL, and BDNF, which regulate intestinal neurotransmitters and brain-derived neurotrophic factors. These findings provide new targets for the treatment of constipation, and also lay a foundation for further research on intestinal neural regulation mechanisms.

In order to further reveal the underlying mechanism of BC99 for constipation relief, serum metabolomics analysis was performed on the placebo and BC99 groups to investigate alterations in the metabolic status of participants with constipation at the 8th week. After BC99 intervention, the abundances of 93 serum metabolites were changed, of which 82 were up-regulated and 11 were down-regulated. It is particularly noteworthy that four metabolites classified as sphingolipids, including SM(d18:1/17:0), SM(d18:1/18:0), SM (d18:0/16:0), and palmitoyl sphingomyelin, showed significant increases after BC99 treatment. KEGG enrichment analysis indicated that BC99 supplementation exerted significant impacts on the sphingolipid signaling pathway, steroid hormone biosynthesis, sphingolipid metabolism, and glycerophospholipid metabolism, which was in agreement with the increased levels of neurotransmitters and hormones in the BC99 group. Additionally, the correlations between key differential metabolites and clinical indicators of constipation were analyzed. As presented in Figure 8, four metabolites involved in sphingolipid metabolism, including SM(d18:1/17:0), SM(d18:1/18:0), SM(d17:1/24:1(15Z)), and SM(d18:0/16:0), were positively correlated with SBM, PCA-QOL, 5-HT, AChE, MTL, BDNF, and anti-inflammatory factors (IL-4 and IL-10), and negatively correlated with the pro-inflammatory factor IL-6. These results suggest that the alleviation of constipation mediated by BC99 is closely correlated with sphingolipid metabolism.

Studies have found that there is a certain correlation between constipation and sphingomyelin metabolism [39]. Sphingomyelin is a sphingolipid containing choline phosphate, which mainly exists on the cell membrane, and participates in life activities such as cell signal transduction and cell apoptosis [40]. It has been reported that sphingomyelin plays an important role in intestinal health and immune homeostasis. Its metabolic abnormalities may affect intestinal barrier function, intestinal smooth muscle contraction, intestinal permeability, and inflammatory responses, thus resulting in the occurrence and development of constipation and other intestinal diseases [41]. Under normal circumstances, the intestinal flora participates in the metabolism of sphingomyelin, breaking it down into sphingosine and fatty acids, which have the function of regulating intestinal smooth muscle contraction [42]. However, the structure of the intestinal flora in constipated patients changes, with a decrease in beneficial bacteria and an increase in harmful bacteria, which blocks the sphingomyelin metabolism pathway and affects intestinal function [43]. This study suggests that the relief that BC99 provides from dysfunction in patients with chronic constipation may be related to the regulation of sphingomyelin metabolism. Further studies regarding the gut microbiota and short-chain fatty acids remain to be conducted to comprehensively clarify the underlying mechanism of BC99 for constipation treatment.

## 5. Conclusions

In this study, the 8-week *W. coagulans* BC99 intervention significantly improved constipation symptoms, enhanced the quality of life, increased the levels of neurotransmitters (5-HT, MTL, AChE, and BDNF) positively associated with intestinal motility, and declined inflammation levels in adults with constipation. In addition, *W. coagulans* BC99 intervention resulted in 93 differential metabolites and influenced several metabolic pathways critical to intestinal motility, especially sphingolipid metabolism. Our findings prove the effectiveness of *W. coagulans* BC99 in the clinical remission of constipation, highlighting its potential application as an effective strategy for the clinical treatment of constipation in adults. Further in-depth explorations are required to comprehensively reveal the mode of action of *W. coagulans* BC99 for constipation relief.

## Figures and Tables

**Figure 1 foods-14-00654-f001:**
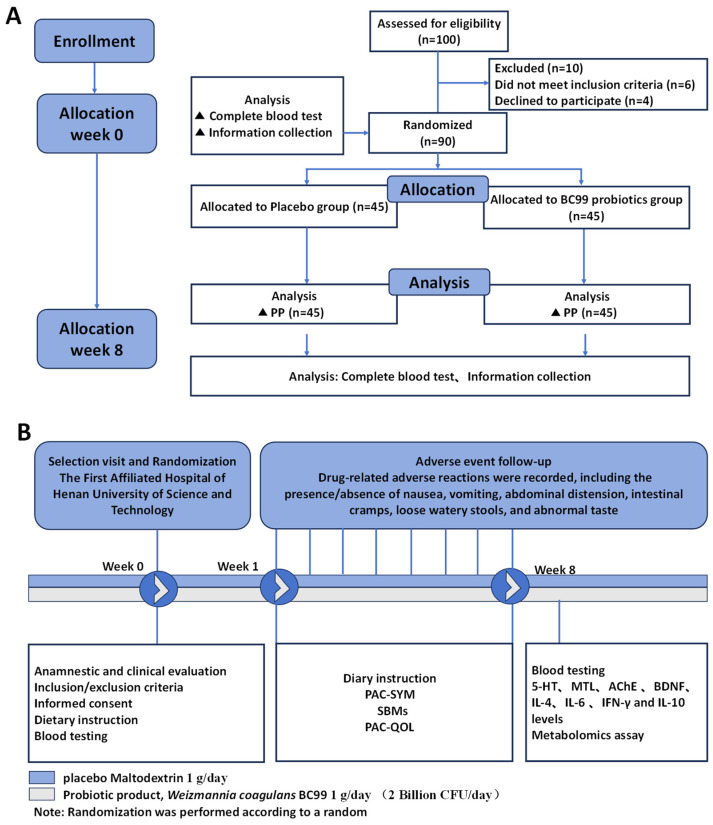
Flowchart for this study. Process of the study implementation. (**A**) The design of the study. (**B**) Flow diagram of this study selection.

**Figure 2 foods-14-00654-f002:**
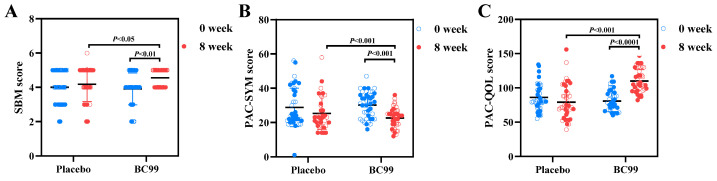
Changes in SBM, PAC-SYM, and PAC-QOL scores after 8-week intervention with BC99. (**A**) SBM scores of placebo and BC99 groups. (**B**) PAC-SYM scores of placebo and BC99 groups. (**C**) PAC-QOL scores of placebo and BC99 groups.

**Figure 3 foods-14-00654-f003:**
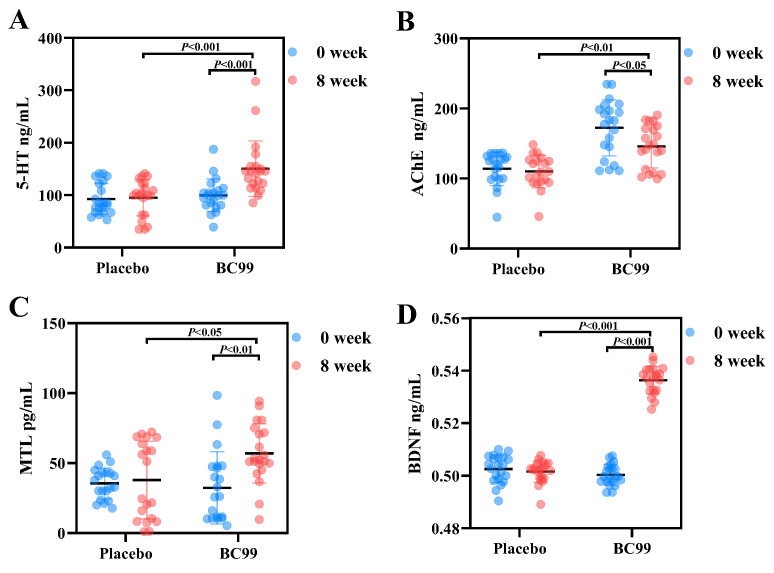
Effects of BC99 intervention on neurotransmitter and hormone levels in patients with constipation. (**A**) 5-HT. (**B**) AChE. (**C**) MTL. (**D**) BDNF.

**Figure 4 foods-14-00654-f004:**
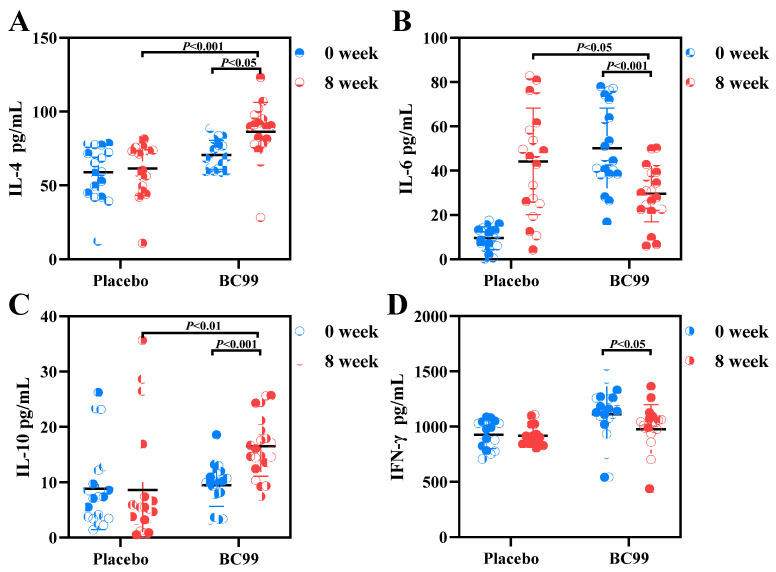
Effects of BC99 intervention on inflammatory factor levels in patients with constipation. (**A**) IL-4. (**B**) IL-6. (**C**) IL-10. (**D**) IFN-γ.

**Figure 5 foods-14-00654-f005:**
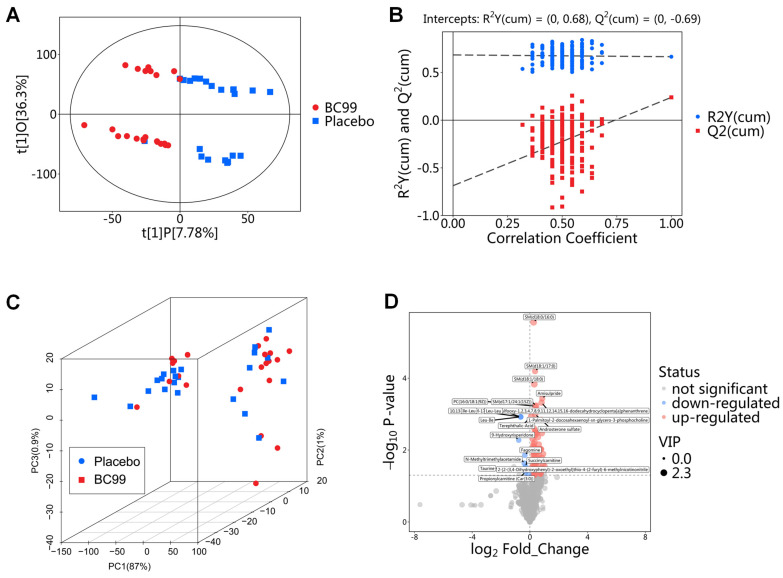
Effect of BC99 on serum metabolic characteristics in adults with chronic constipation at 8th week. (**A**) OPLS−DA score plot. (**B**) OPLS−DA permutation test. (**C**) PCA score plot. (**D**) Volcano diagram.

**Figure 6 foods-14-00654-f006:**
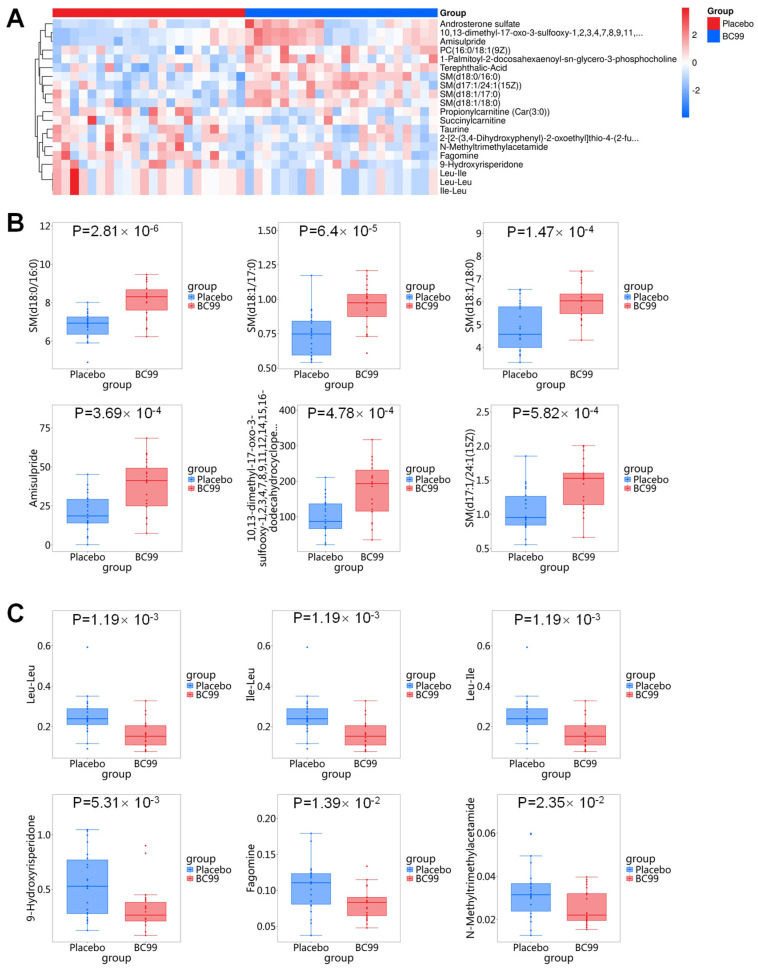
Effects of BC99 intervention on serum metabolites in adults with chronic constipation at 8th week. (**A**) Heat map of differential metabolites. (**B**) Relative abundance of up-regulated metabolites. (**C**) Relative abundance of down-regulated metabolites.

**Figure 7 foods-14-00654-f007:**
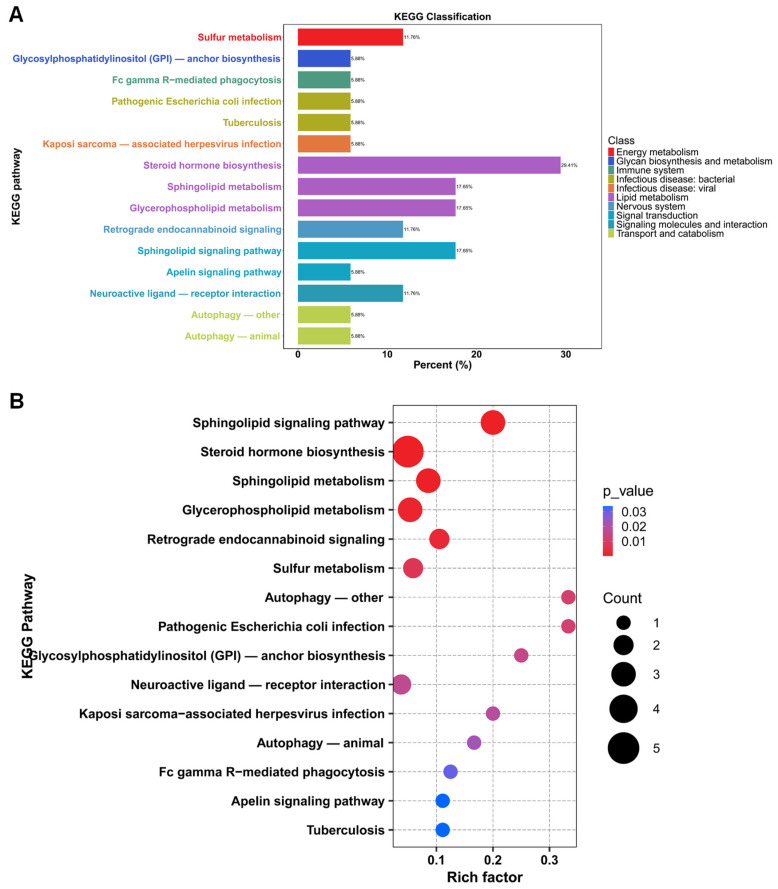
(**A**) KEGG classification of differential metabolites between placebo group and BC99 group. (**B**) KEGG enrichment analysis of differential metabolites between placebo group and BC99 group.

**Figure 8 foods-14-00654-f008:**
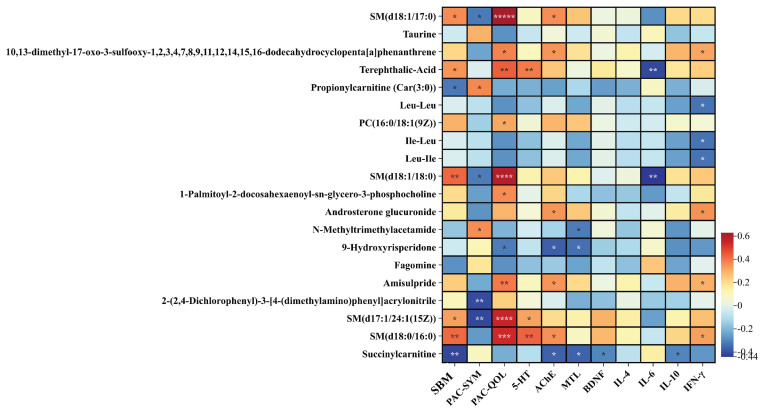
Correlation analysis between serum important metabolites and clinical indicators. Blue represents negative correlation and red represents positive correlation. * *p* < 0.05, ** *p* < 0.01, *** *p* < 0.001, **** *p* < 0.0001, ***** *p* < 0.00001.

## Data Availability

The original contributions presented in the study are included in the article/Appendix A. Further inquiries can be directed to the corresponding author.

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
