# Peer review of "Weizmannia coagulans* BC99 Relieves Constipation Symptoms by Regulating Inflammatory, Neurotransmitter, and Lipid Metabolic Pathways: A Randomized, Double-Blind, Placebo-Controlled Trial"

_foods, 2025, doi:10.3390/foods14040654_

Round 1

Reviewer 1 Report

Comments and Suggestions for Authors

Comments to Manuscrip (Qiuxia Fan et al., Foods, ID:foods-3434325) “Weizmannella coagulans BC99 relieves constipation symptoms by regulating inflammatory, neurotransmitter and lipid metabolic pathways: a randomized, double-blind, placebo-controlled trial.”

Fan Q. et al demonstrate that oral intake of probiotic bacteria Weizmannella coagulans BC99 significantly improved constipation symptoms as compared with placebo-treated group in clinical study. The authors further reported  the mechanisms of BC99 supplementation in alleviating constipation, such as increase of neurotransmitters and higher levels of anti-inflammatory cytokines. This article contains attractive topics for many readers in the related fields, and can be published after some revisions described below.

Major comments

1.      In section 2.3., is BC99 commercially availiable probiotic in China?  How is its daily dose determined to 1x10^10 CFU/day? And when did the participants take probiotic/placebo (after supper, or after every meal)?

2.      In section 2.6., Line 154-155, does the word “SBMs” mean complete spontaneous bowel movement throughout this manuscript? Is the “CSBMs” (Line 355-356, Figure 8) same as “ SBMs score” in Figure 2?

3.      In Figure 2, what does mean the word “change” (in vertical axis)?  Ratio or difference between before and after the each intervention?  Data are expressed as means±SD.  But I think it better to use dot type graph like as Figure 3, because distribution of each subjects and significant difference can be easily understanded.

4.      In section 3.1., before the intervention, was there no difference in constipation level between probiotic group and placebo group?

5.      In Figure 3B, the authors indicated significant increase of AChE compared with the placebo group. However, AChE level tended to be higher even at 0 week in probiotic group compared with 0 week placebo group, and decreased significantly at 8 week after intervention. While the authors mentioned that BC99 intervention can affect intestinal motility by increasing the levels of 5-HT, AChE, etc (Line 368-370), more interpretation about AChE level should be added.

6.      In Figure 4BD, the authors indicated significant decrease of inflammatory cytokines by probiotic intervention. However, IL-6 level ,compared with 0 week in placebo group, seems to be higher at 8 week in placebo group, and also higher at 0 week in probiotic group. How do the authors expain this point?  

Minor comments

1.      Line 50, I recommend the authors to add one or two sentence explaining the importance of gut-microbiota and gut-brain interaction in GI disorders. Ref 1 and 2 can be used as instructive references for readers.

2.      Line 60, “Comapred with ” may provably be “Compared with  ”..

3.      In Figures, applied statistics method should be written.  If the authors write the p value, asterisk *,**,*** can be deleted.

4.      Line 126, “six possible blocks wi th“ may provably be “six possible blocks with“..

5.      Line 375, “ alternations of metabolic status” may be “alterations”.

6.      In References list, reference numbers 15~25, 27, 29, 31,32, 35, 36 are repeated.

7.      Please check again the form of Table S1. In Table S1, basic indicators of participants at baseline and 8th week (after intervention) are listed. But, “ p-value^2) (intergroup)” may be “p-between between baseline and 8th week data” ( Is that correct?). Please insert some space between the each indicators. And some explanation about this supplemental data should be added in Results or Discussion section.  

8.      Table S2 might summarize the metabolomics data of 93 changed serum metabolites describe in Line 376. Are the values in the Placebo and BC99 column average of metabolites concentration? If so, add the concentration unit, please. And some explanation about this supplemental data should be added in Results or Discussion section.  

Reviewer 2 Report

Comments and Suggestions for Authors

Here are my comments: 

1. Please insert more references on page 2 lines 53-54. 

2. May you please go through the manuscript and rectify grammatical and typos. 

3. In Figure 2, as the SD is high have you considered using the non-parametric tests like the Mann-Whitney U test or the Kruskal-Wallis test. 

4. In Figure 3 D, the probiotic color used for 0 weeks and 8 weeks are different then rst of the figures in Figure 3 block. 

5. In Figure 4 A, the probiotic group color needs to be corrected as per other images enclosed in figure 4. 

6. May you insert reference on page 13 line 394.

7. Did you notice any change in the abundance of Desulfovibrio, Coprococcus with the BC99 treated group? 

Comments on the Quality of English Language

There are numerous typos and grammatical errors throughout the manuscript, that need to be rectified. 

Reviewer 3 Report

Comments and Suggestions for Authors

A brief summary:

Aim of this study was to examine probiotic effect of Weizmannella coagulans BC99 on constipation symptoms in randomized placebo-controlled trial for 8 weeks. The spontaneous bowel movement frequency (SBMs), patient assessment of constipation symptom (PAC-SYM), patient assessment of constipation quality of life (PAC-21 QOL), inflammatory cytokines, neurotransmitters, and serum metabolites were investigated with the aim of discovering a possible mechanism of action. Results showed that BC99 increases bowel movements frequency, relieves constipation symptoms (increased CSBM with increase PAC-SZM and PCA-QOL scores), affects levels of pro-inflammatory and anti-inflammatory cytokines (elevated levels of IL-6 and IFN-, increased levels of IL-4 and IL-10) and increases the levels of 5-HT, AChE, MTL and BDNF neurotransmitters. Additionally, it seems that mechanism of action of BC99 includes sphingolipid metabolic pathways.

General concept comments:

The Manuscript is well written and the coverage of the topic is also good, but some minor additions should be included:

1. Some additional references should be added in the Discussion section. Cytokines are hardly discussed at all (just mentioned in line 388). Additional references are necessary in order to better understand possible mechanism of action of BC99 through neurotransmitters and metabolic pathways. What has been shown so far in other studies and how it can be connected with your obtained results (beside these 28, 29, 31, 32, 33, 34, 35, 36). It is not necessary to describe the results again in detail in the discussion.

2. Weizmannia coagulans BC99 (BC99) is a Gram-positive, spore-forming and lactic-acid-producing probiotic strain isolated from the fecal sample (Zhu, M.; Zhu, J.; Fang, S.; Zhao, B. Complete genome sequence of Heyndrickxia (Bacillus) coagulans BC99 isolated from a fecal sample of a healthy infant. Microbiol. Resour. Ann. 2024, 13, e00449-23.).

Since you are examine effect of bacteria that has a probiotic effect, have been stool samples analyzed, gut microbiota composition?

For example: Fecal samples were analyzed for moisture content, short-chain fatty acids, branched-chain fatty acids, microbiota composition, and calprotectin….” (Cheng J, Gao C, Ala-Jaakkola R, Forssten SD, Saarinen M, Hibberd A, Ouwehand AC, Ibarra A, Li D, Nordlund A, Wang Y, Shen X, Peng H, Wan X, Meng X. Eight-Week Supplementation With Bifidobacterium lactis HN019 and Functional Constipation: A Randomized Clinical Trial. JAMA Netw Open. 2024 Oct 1;7(10):e2436888. doi: 10.1001/jamanetworkopen.2024.36888. PMID: 39356506; PMCID: PMC11447574.)

Specific comments:

1. Spelling check: line 60, “compared”, line 80, “(1×1010 CFU/day” add bracket, line 515 “36.

36. Zhao, X.; Wang…” twice numbered

2. Line 365-366: please adjust font size

Reviewer 4 Report

Comments and Suggestions for Authors

All comments, suggestion, and questions are available in the manuscript.

Reviewer 5 Report

Comments and Suggestions for Authors

Comments are in pdf. Very clearly written, would only improve the resolution where the graphics are presented, very difficult to follow.
